# Urban Governance of Household Waste and Sustainable Development in Sub-Saharan Africa: A Study from Yaoundé (Cameroon)

Salifou Ndam [1,*], Alirou Fit Touikoue [2], Jérôme Chenal [1], Jean-Claude Baraka Munyaka [1], Armel Kemajou [1] and Abdou Kouomoun [2]

[1] School of Architecture, Civil and Environmental Engineering, École Polytechnique Fédérale de Lausanne, 1015 Lausanne, Switzerland; jerome.chenal@epfl.ch (J.C.); baraka.munyaka@epfl.ch (J.-C.B.M.); armel.kemajou@epfl.ch (A.K.)

[2] Department of Sociology, Faculty of Arts, Humanities and Social Sciences, University of Yaoundé 1, Yaoundé P.O. Box 755, Cameroon; abdoukouomoun@gmail.com (A.K.)

[*] Correspondence: salifou.ndam@epfl.ch

**Abstract:** More and more cities in Southern Africa are struggling to manage their waste in a context of rapid urbanisation and increasing poverty. In the Cameroon's largest city, Yaoundé, managing household waste is a growing concern. The public and the authorities cast blame on each other, and the actions taken by each party far from guarantee an efficient management of household waste, which litters the streets. Considering the above, this paper analyses the socio-political practices of stakeholders and their influence on household waste management in Yaoundé. Based on a qualitative survey that combined both a literature review and interviews, the research analysed the challenges related to household waste management with regard to the economy, the environment, and public health. In addition, a cartographic survey using KoboToolbox was conducted in all seven municipalities to analyse the geographical distribution of the waste areas, their size, and their status within the city organizational framework. In total, 264 waste dumps were collected, of which 110 were formal waste using a waste bin of varying size. Social constructivism, stakeholder theory, and strategic analysis were mobilized to analyse the urban waste governance in Yaoundé. Thus, the poor quality of household waste management in Yaoundé was explained using political, economic, sociocultural, and environmental parameters. The social practices and dynamics of the stakeholders generate undesirable consequences that hinder the achievement of the Sustainable Development Goals (SDGs). By combining social science and engineering methods, this research aims to demonstrate that the shortcomings of waste governance in Yaoundé are both a collective (authorities/public powers) and individual (citizens) matter.

**Keywords:** household waste; waste management; urban governance; sustainable development; Yaoundé; Cameroon

## 1. Introduction

The concept of sustainable development is an integral part of the scientific, political, economic, and socio-cultural discourse in both industrialised and developing countries. Since the Brundtland principles called for reconciling economic growth, environmental protection, and social equality, more than 193 countries have embraced sustainable development policies [1]. These elements are presented as a logical framework for the future of societies. Almost all cities in developing countries (DCs) have faced household waste management problems in recent decades. Waste management is the process of collecting, transporting, treating, reusing, or disposing of waste to reduce its impact on human health and the environment. Household waste is "all putrescible or non-putrescible materials of animal or plant origin resulting from food handling, preparation or consumption" [2].

Urban sprawl and population growth have yet to be accompanied by the necessary means to ensure sustainable development in this area. Governance, an all-encompassing social reality, is how we can and must guide the course of society [3]. Urban governance is a framework for reflection that aims to make the processes that impact the governability of cities intelligible. The problem of household waste management in Africa is worrying, whether in large or small cities [4,5]. However, the situation is even more pronounced in sub-Saharan Africa, which, for example, in 2012 produced 81 million (65%) of the 125 million tonnes of waste produced by the whole continent [6]. With an inconsequential annual increase, this is expected to rise to around 244 million tonnes per year by 2025 [5]. However, the management methods implemented by the authorities in many cities (countries) need improvement to produce the desired effects [7,8].

The capital of Cameroon (a country located in Central Africa), Yaoundé, is the country's main city, with a population of 4 million. Although not the largest city in Africa, it has experienced in recent years strong and steady population growth since the country's independence in 1960 [9]. The increasing population complicates the existing efforts towards household waste management. Today, Yaounde contributes to the 12.6 million yearly deaths the World Health Organization (WHO) has reported as being caused by environmental insalubrity [10]. This figure is expected to rise dramatically in the coming years due to the growing urbanisation in Sub-Saharan African countries [11]. Hence, Marcel Mauss considers open waste deposits to be among the most important things to study in society [12]. This is all the more important when decision makers are unable to manage waste [13]. In recent decades, household waste management has increasingly been regarded as an environmental health issue [14]. The sociological impact of the crisis on sustainable development, although not often mentioned, is worth further analysis. The interest also extends to the socio-anthropological, targeting the influence of household waste on the environment, public health, and the urban economy.

As stated by Cameroonian long-serving president Paul Biya (since 1982): "As long as Yaoundé breathes, Cameroon lives" [15]. However, due to the severe sanitation problems observed and the current inability of local authorities to tackle the crises, the social, political and economic repercussions are felt even beyond Yaoundé boundaries. The annual bonus for excellence in sanitation granted by the government under the "Clean Cities Programme" launched in 2019 has not had the expected effects on city hygiene for several reasons, including (i) the lack of planning to assess the real need of local authorities, (ii) the absence of a monitoring and evaluation systems, (iii) the small amount of money available, (iv) and the absence of trained personnel who can reform local practices.

On the environmental level, Epoh-Mvaboum and Essomba Ebela [9–12] considers household waste management as the origin of the environmental degradation in Yaoundé. Meanwhile, Belomo Mvondo [16] points out the consequences of household waste management on the environment and public health and argues that Yaoundé contradicts the requirements of sustainable development. In examining the correlation between sustainable development and household waste management, Tini [17] believes that sustainable development is a means that opens up new perspectives for cities. This is a dynamic of action oriented towards realizing long-term projects defined by each local authority and guided by renewed principles of action.

On a socio-economic level, Gagoa Tchoko and Michael [18,19] focused their analyses on the economic opportunities the household waste management sector offers for sustainable development. Ngnikam [20] conducted a qualitative and quantitative assessment of household waste management in Yaoundé and noted that the quality of service provided in terms of sanitation by public and private actors is mediocre. On an anthropological level, Monsaingeon [21] analyses the link between household waste management, sociocultural representations and religious practices. Zoa [22] considers that society reveals itself through its waste, as its production, management, and exploitation reflect the aspirations of a community. In addition, the World Bank [23] draws the attention of developing countries to the impacts of household waste management and proposes a practical solution, namely

popular participation in environmental management. In addition, it calls for inclusive waste management for sustainable and efficient household waste management. Although important, this proposal is limited, as it focuses exclusively on the ecological dimension of household waste management. This article outlines the main dimensions of sustainable development, notably social, economic, and environmental, and focuses on how social actors construct their relationships with waste.

From a theoretical standpoint, contrary to other works that have used the theory of economic management [24], cognitive approaches [18] or the idea of social representations [25], this research utilizes three different theories: (i) the theory of social constructivism (which links social structures and cognitive structures to explain the decisions of public authorities and the choice of waste disposal sites by the population) [26], (ii) the stakeholder theory (which considers the responsibilities of public authorities and local populations in the waste management crisis) [27], and (iii) strategic analysis (which allows us to study the strategies developed by local people to dispose of their waste in the city) [28]. Although most research on waste is unanimous in the opinion that Africa is languishing under the weight of its waste, there has yet to be any (known) research that visualizes the scale of the waste management problem at the city level. By combining social science and engineering methods, this research aims to demonstrate that the shortcomings of waste governance in Yaoundé are both a collective (authorities) and individual (citizens) matter.

## 2. Materials and Methods

### 2.1. Data Collection Process

Data were collected in three complementary stages (Figure 1). The first stage took place from January to August 2022. In this stage, the exploratory interviews were first conducted with 70 users/individuals (10 per commune) with diverse profiles (civil servants, health workers, street traders, students, etc.). The discussions focused on (i) the difficulties encountered in household waste management, (ii) the quality of service offered by institutional and private actors, (iii) the performance of waste collection agents, and (iv) the satisfaction of the population with household waste management. In parallel, regular direct observations were made in the streets (central and secondary) of the different neighbourhoods, with the aim being to determine the population's relationship with waste. In addition, a literature search was carried out online (social media, databases, etc.) and in local libraries in Yaoundé to gather the necessary resources related to waste management.

The second stage took place between November 2022 and January 2023. During this period, direct observations continued, and in-depth interviews were conducted with a snowball sampling of two categories of social actors: (i) institutional or public actors and (ii) private actors or service concessionaires in charge of waste management. Thus, a total of 33 interviews were conducted, 19 with women and 14 with men. Based on the two main categories of interviewees, two interview guides were elaborated to ensure that the discussions were focused on questions specific to each of the fields of intervention. Some of the questions raised include: How much waste do you produce daily? Where do you dispose of the waste produced or collected? Is it important for you to separate trash (why)? Is it possible for districts to regulate the amount of waste households produce? What impact does waste dumped on the streets have on your health and the environment? What direct action are you taking to ensure sustainable waste management? What are companies doing to reduce packaging? What happens to the waste collected (or dumped) on the streets?

As far as institutional actors are concerned, the main questions addressed comprised (i) the legal, regulatory, and legislative frameworks and their application, (ii) the current state of household waste management, and (iii) the effective consideration of sustainable development. The executives surveyed work at the Ministry of Housing and Urban Development (MINHDU), the Yaoundé City Council (YCC), and the 7 Yaoundé District Municipalities (YDMs), among others. The YCC is responsible for the overall management (directed by a "super" City Mayor), while the 7 YDMs directed by a mayor are responsible for local management (Figure 2).

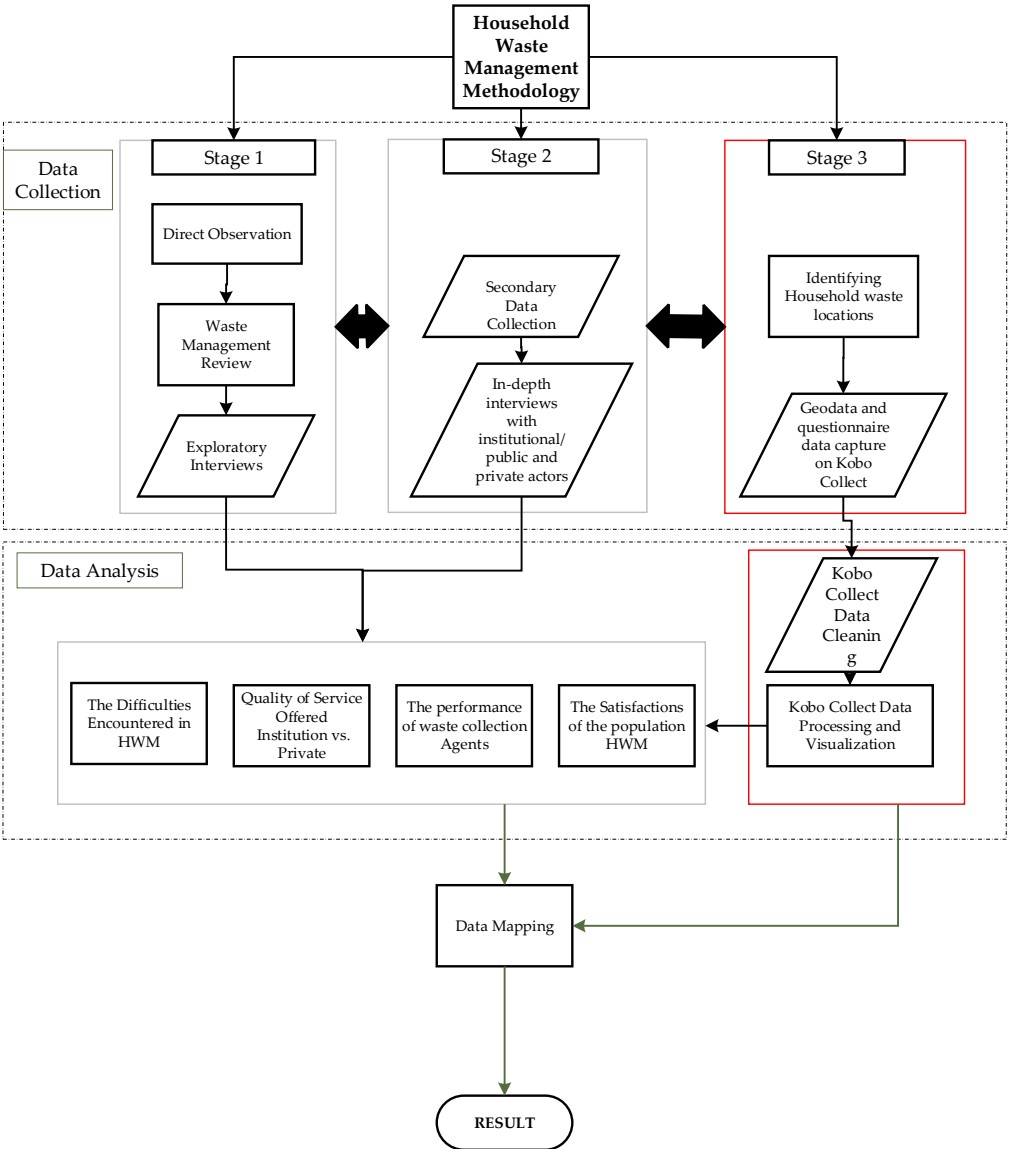

**Figure 1.** Data collection and analysis process. The red boxes represent key moments in the re-search process.

As for the private actors (or concessionary companies) interviewed, the following were included: (i) the communication officer of HYSACAM (the leading legal company in charge of street waste collection and treatment), (ii) HYSACAM employees (garbage collectors, mobile collection agents, etc.), (iii) civil society (leaders and members of the Tamtam-Mobile and Coeur d'Afrique Foundation Roger Mila (CAFROMI) associations, etc.), (iv) members of neighbourhood committees (Biya Youth Association), (v) members of the local community, (vi) pre-collection agents, (vii) waste recovery agents, (viii) ragpickers, (ix) waste re-users, (x) waste transformers, and (xi) sellers of waste thrown into the streets (dumps), etc. The interviews focused on (i) knowledge of the regulatory and legislative texts on household waste management; (ii) the waste collection, transport, and treatment system; and (iii) the implementation of waste management mechanisms and strategies regarding sustainable development.

The third data collection stage occurred during the fourth week of March 2023. Using KoboToolbox, the questionnaire targeted (i) the state of household waste, (ii) waste location, and (iii), if available, the bin containing this waste or the lack thereof. In addition to the questionnaire, the geolocation of the household waste was determined, including spatial

layout and size of formal and informal waste dumps in the streets of the 7 YDMs of Yaoundé and the frequency of removal of piled up waste.

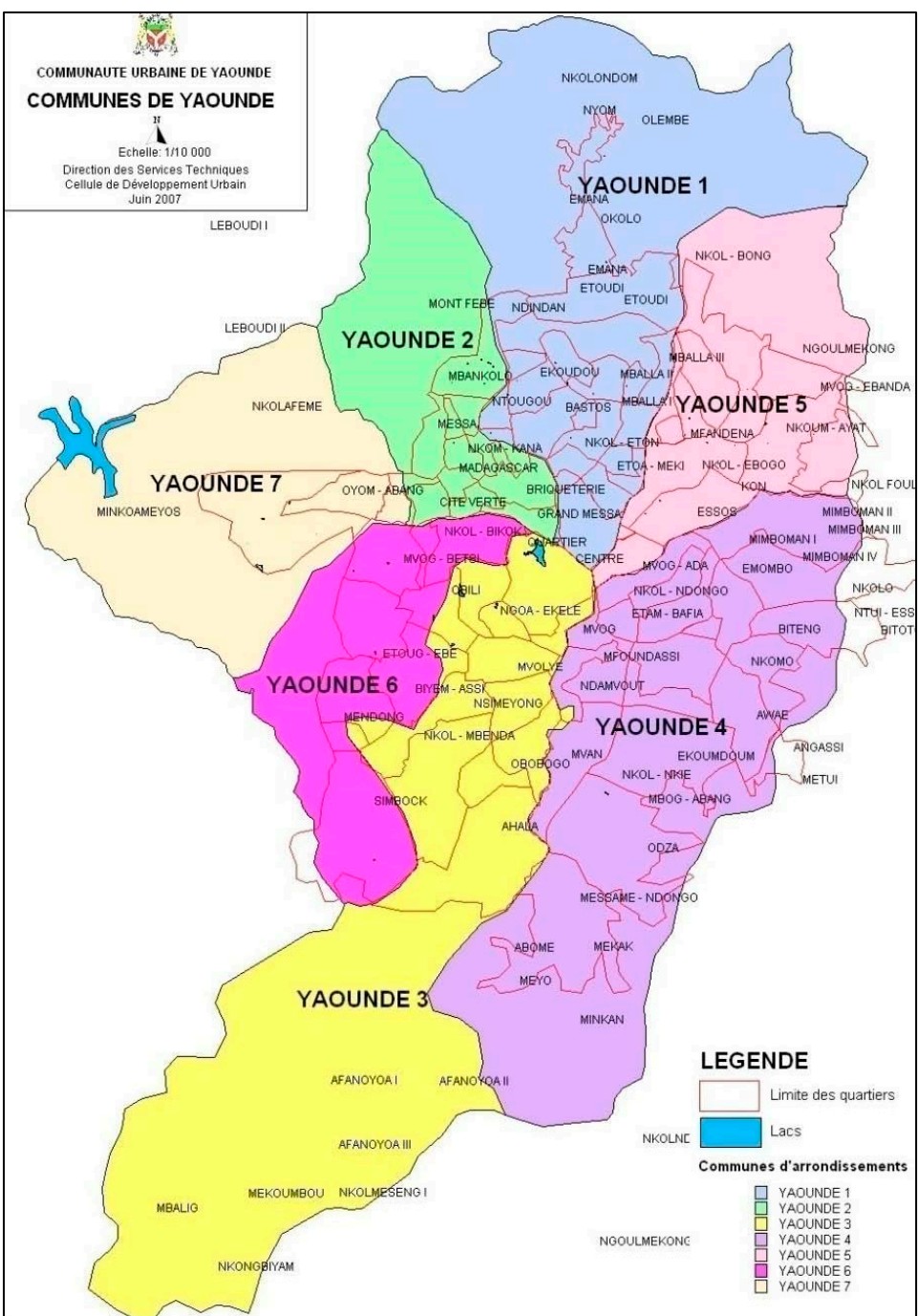

**Figure 2.** Map of Yaoundé, including its 7 districts as well as municipalities and neighbourhoods. Used with permission.

### 2.2. Data Processing

The interviews and direct observations were analysed using thematic content analysis [24]. After transcribing the interviews, semantic units were formed to highlight and combine the themes of the observation notes and photos taken in the streets. Then, data collected through KoboToolbox were analysed using statistical and geospatial means (Table 1). The following categorizations of the listed attributes were taken into account:

**Table 1.** Categories of data collected through KoboToolbox.

| SN | Data Type | Categories |
|---|---|---|
| 1 | Yaoundé district municipalities (YDMs) | Yaoundé 1, Yaoundé 2, Yaoundé 3, Yaoundé 4, Yaoundé 5, Yaoundé 6, and Yaoundé 7 |
| 2 | Formality of the household waste locations | Formal or Informal |
| 3 | Waste bin presence | Yes or No |
| 4 | Bin size | Large (16–30 m$^3$), Medium (7–16 m$^3$), and Small (0–6 m$^3$) |
| 5 | Waste area size | Between 10 and 50 m$^2$, below 10 m$^2$, above 50 m$^2$ |
| 6 | Frequency of waste bin being emptied | Weekly, Daily, Monthly, Never |
| 7 | Condition of the waste bins | Halfway Full, Full to Capacity, Empty, Overflowing |
| 8 | Quality of routes reaching the bin areas | Asphalt route of good quality, Non-asphalt route of good quality, Asphalt route of bad quality, Non-asphalt route of bad quality |
| 9 | Accessibility of the bin areas | Bicycles, Motorcycles, Tricycles, Trucks, Cars, Walking |

From the listed data collected and the categories, interpretative analysis and maps were generated using Quantum Geographic Information System (QGIS).

## 3. Results

### 3.1. Stakeholders and Household Waste Management in Yaoundé

Studies carried out from 2000 onwards in African cities show that the strategies adopted by states to manage DM are not very effective [19–22]. To strengthen its waste management policy, the State of Cameroon has ratified several international conventions (Table A1). To be effective, it has established a legal framework and institutions deemed competent (Table A2). Specific roles have been assigned to central services and decentralised local authorities (Table A3).

To this end, there are many institutional actors directly involved in household waste management, which can lead to confusion. The Ministry of Housing and Urban Development is responsible for implementing the national policy on housing and urban development and for monitoring the application of hygiene and sanitation regulations as well as the removal and treatment of household waste. The Ministry of Territorial Administration and Decentralisation is responsible for (i) raising awareness of households in the neighbourhoods; (ii) repercussions in case of infractions; (iii) financial contribution for waste removal; and (iv) removal and treatment of household waste. The Ministry of Environment, Nature Protection and Sustainable Development is responsible for (i) the elaboration of a National Environmental Management Plan; (ii) the examination of files relating to the elimination, recycling, and burial of waste in liaison with the administrations concerned; and (iii) the periodic control of waste landfills. The Ministry of Public Health prescribes conditions for removing and treating urban waste. The Ministry of Agriculture and Rural Development is responsible for raising awareness of the population on transforming putrescible waste into fertiliser.

The city councils (such as YCC) are responsible for enforcing hygiene measures, sanitation, awareness-raising among the population, and collecting the direct tax on household waste removal. The Yaoundé district municipalities (YDMs) are responsible for ensuring sanitary, local, and environmental policing using sanitary engineering agents. Civil society organisations (NGOs, neighbourhood associations, etc.) support the development of widespread ecology and sanitation awareness movements and projects. In addition, private companies and service concessionaires include (1) HYSACAM (Hygiene and Health in Cameroon), which is responsible for (i) raising awareness among the population to prevent the dumping of waste on the streets and (ii) collecting, transporting, and disposing of waste in the cities; and (2) Common Initiative Groups (GICs) and (3) Small and Medium Enterprises (SMEs), which are private services that pay for the pre-collection of waste.

Finally, users contribute to financing household waste collection through the hygiene and sanitation tax, and local waste collection associations also contribute to waste collection. Despite this array of actors officially involved in waste management, informal and unofficial waste disposal sites are still highly prevalent in the streets of Yaoundé. The HW data were collected in all 7 Yaoundé district municipalities (YDMs), targeting 264 disposal sites. Figure 3 shows that 58.33% of the collected disposal sites are informal, also called "wild" waste. Figure 3 also shows the geolocation of both informal and formal areas. These are the result of the deliberate choice of citizens to deposit their waste at any location in the street regardless of whether HYSACAM will be able to collect it.

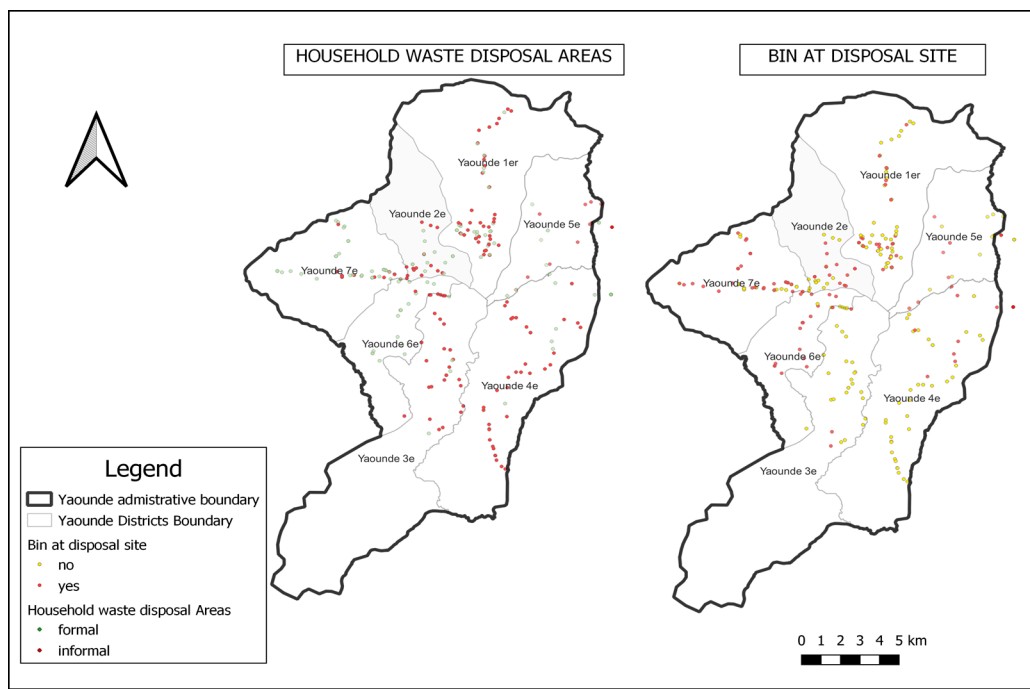

**Figure 3.** Geolocation of waste disposal sites with or without waste bins in the city of Yaoundé.

Of the 264 waste deposit sites recorded, only 110 (41.67%) were formal waste deposit sites registered by the YCC and with a daily collection schedule by HYSACAM. Figure 3 above also shows that among the formal waste deposit sites collected, only one site did not have a bin for waste collection. Bins used in Yaoundé are mostly steel or plastic waste containers with varying load capacities and are provided for this purpose. Bins are categorized into three sizes, Large bins are between 16 and 30 m$^3$, and medium and small bins are between 7 and 15 m$^3$, and 0 and 6 m$^3$, respectively. Figures 4 and 5 below show the percentage of bin sizes found on-site and their geolocation, respectively.

Figure 4 shows that most refuse bins are large (45%), followed by medium (29%) and small (26%). It should be noted, however, that, due to planning matters (no regular collection as well as limited number of bins in highly densified areas), all refuse bins surveyed were overflowing with refuse. The city's distribution of large refuse bins is disproportionate to the different waste collection points. The assumption is that this disparity is due to the lack of geospatial planning for bin location as well as the limited involvement of district authorities in the maintenance of public health; this is also a function of the differences in financial resources available between the districts.

Like all districts in Cameroon, the seven districts of Yaoundé are a decentralised territorial authority, acting as independent entities and capable of making their own decisions under the supervision of the central State [29]. Districts in Yaoundé also manage financial the resources generated in-house as well as the additional funds given as a district yearly budgeted allocation. The central state funds, also Called "General Operating Allowance", are allocated to contribute to "good functioning" and investment (road maintenance, elec-

trification, etc.). In January 2023, for example, all 360 districts and 14 City Councils of Cameroon received XAF 128,108,015,786 (EUR 195,299,389) from the central government. Unfortunately, this contribution is insignificant compared to Sub-Saharan African states well known for their cleanness, such as Rwanda (e.g., Kigali) or Senegal (e.g., Dakar), which have better waste management. Regarding investments, South Africa invests far more than any African country [30]. Kigali has a low budget but boasts efficiency [31]. In principle, the financial allocations in Cameroon are distributed in proportion to the district population size and should be managed by each district on behalf of the whole population. In Yaoundé, however, the allocation is mostly managed by district authorities based on political affiliation or the allegiance to the governing political party. Although, officially, the seven districts of Yaoundé are equal, there is a disparity in the distribution of financial resources between districts. Moreover, a study conducted in Lausanne (Switzerland) shows that in the context of urbanisation, municipal solid waste generation stabilises as incomes increase [32].

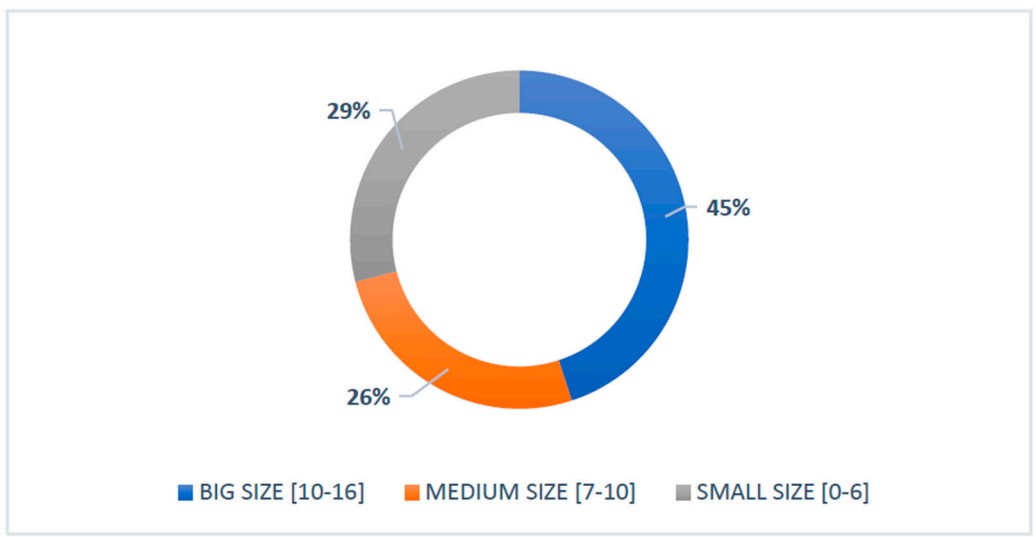

**Figure 4.** Different sizes of formal and informal waste deposits along the streets.

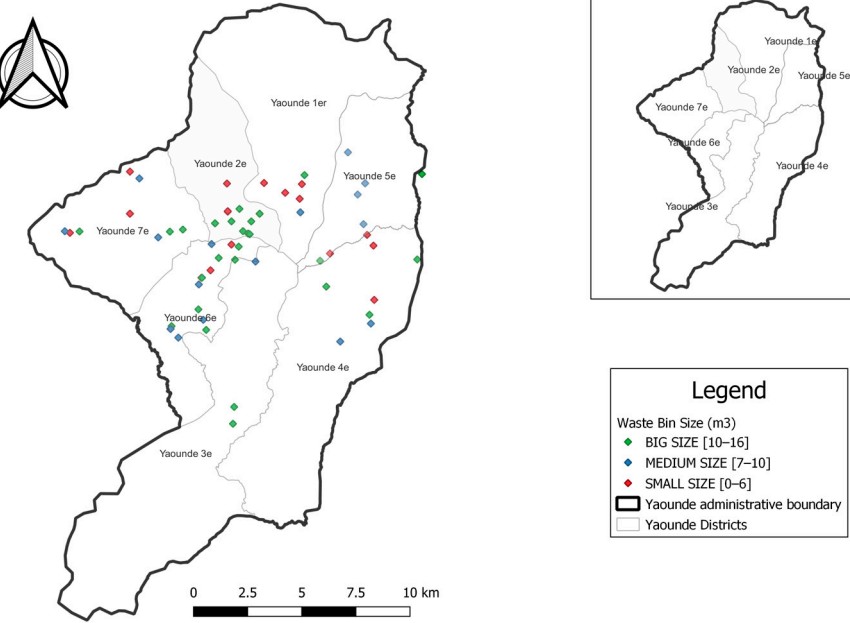

**Figure 5.** Geographical distribution of the different sizes of formal and informal waste deposits along the streets.

### 3.2. Inadequacy of Standards and the Gap in Compliance

Like most African cities, Ngnikam and Tanawa [12] and Boukar [28] believe that Yaoundé lacks specific appropriate legal frameworks for household waste management [12,28]. Some of the current decrees in force structuring waste management are over 30 years old, further delaying the re-launch of activities as well as institutional and technical reorganisation [14]. With no decrees enforcing the above laws, their real applications are limited, regardless of the laws themselves. The current texts have shown their limits, especially regarding "demographic pressure observed in Yaounde good management of household waste" [33]. For example, Law No. 96/12 of 5 August 1996 regarding the environmental management framework considers the 'polluter pays' principle in Article 9. However, this provision cannot yet be applied in 2023 because no implementing ordinances allow polluters to pay taxes related to their negative environmental impacts. A YCC executive certifies this: "Some texts suffer from the lack of their application following the example of the Polluter-Pays Principle, which is not very often respected, and whose elaborated sanctions are not applied" [34].

The other parameter explaining the insalubrity in the city is the laxity of the city authorities. The latter seem excessively indulgent towards the deviant practices of the populations. To illustrate this carelessness on the part of the city authorities, articles R367, R369, and R370 of the Penal Code provide for a penalty ranging from XAF 200 to 3600 (EUR 0.30 to 5.49) for anyone who contravenes legal decisions relating to household waste management, such as dumping waste in the streets and waterways, creating illegal dumpsites in neighbourhoods, or incinerating waste in the open. In Biyem-Assi, for example, a dump was made in a cemetery, and waste is regularly incinerated in the open in full view of others. According to a resident of this neighbourhood: "Only God knows if HYSACAM will ever come to this no-man's land again. This other waste pile has been here for at least three months; the road is blocked. It's not easy to get anyone to pass, even on a motorbike or car; you have to pass quickly, sealing your nostrils and mouth. When it passes us, we set it on fire!" [35]. The behaviour of city dwellers in Yaoundé explains the level of application of regulations. The interactions that individuals establish with their living environment also depend on the standards imposed on them and the way they perceive, evaluate, and represent the environment in which they live [36].

Four things are visible (Figure 6) around this waste pile: (i) the receptacle provided by the authorities is overflowing with waste; (ii) the waste occupies a good part of the public road; (iii) traces of leachate are visible on the road; (iv) the waste is in an area with motorcycle taxi drivers and small shops all around. To a certain extent, socio-anthropological logic also constitutes an explanatory parameter of the opaque household waste management in Yaoundé. By analysing social representations and the different modes of waste treatment in West Africa, Watelet [25] shows an ambiguous relationship between societies and the waste they produce. He points out that the meaning that individuals give to waste varies according to time and society, notably according to the ecological, material/economic, or political context, but also according to social conditions and cultural or religious conceptions [25]. Some believe the waste piles observed in public spaces to be the refuge of supernatural forces and the sites of occult practices. Several people interviewed in this study also mentioned the place of ritual practices in maintaining and producing waste in the streets. Furthermore, the HYSACAM official explains that "the identity logics and complex practices of the populations in waste management constitute a reality of the city" [37]. Based on this observation, some researchers, including Diabaté [34] and Ngnikam and Tanawa [20] believe that witchcraft has taken on unprecedented forms in contemporary Africa (e.g., child witches, ritual crimes, etc.) [12,34].

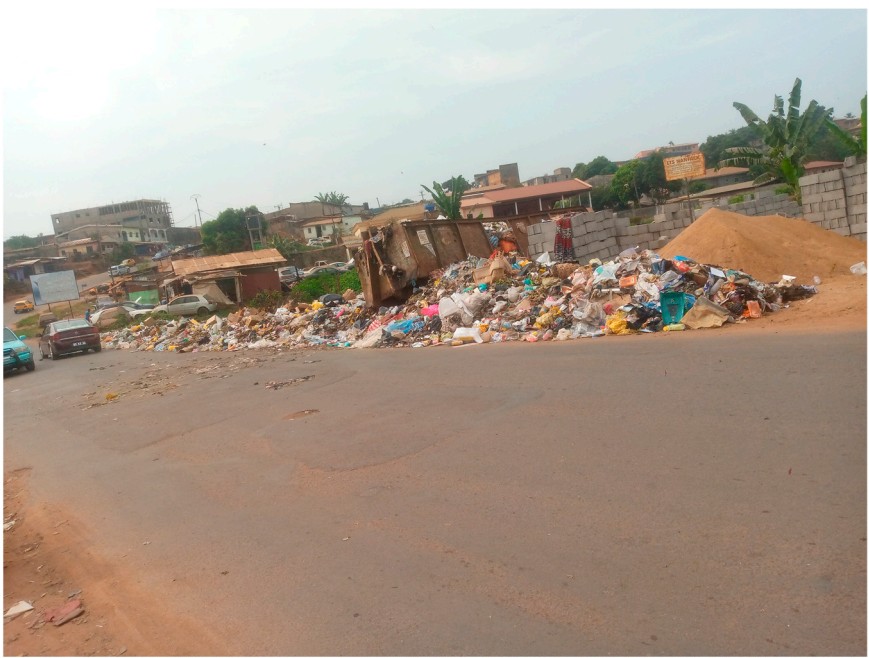

**Figure 6.** Formal waste dumping is well beyond the container's limits at the Nsam junction (YDM 3). Author's field photo, March 2023.

*3.3. The Challenges of Household Waste Management for Sustainable Development*

Yaoundé's climate is conducive to rapidly decomposing household waste in dry or rainy seasons [38]. Its geographical position between $3^e$ and $11^e$ degrees North latitude imposes a Guinean equatorial climate, with abundant rainfall of between 1600 and 2000 mm/year, temperatures of around 24 °C, and humidity between 70 and 80% (in the dry season). As a result, the waste present in certain streets in Yaoundé gives off strong odours, attracting numerous animals and insects (rats, mosquitoes, cockroaches, flies, etc.), which are likely to contaminate human areas and degrade the immediate environment.

According to some inhabitants of Tsinga-City, the waste pile of about 50 $m^2$, as shown in Figure 7, is often there for almost a month and becomes more significant daily. Like many other formal or informal waste piles abandoned by HYSACAM, this pile can lead to verbal and physical violence (brawls, etc.) between those who leave their waste there and those who live in the vicinity. There is even a privatisation of informal waste dumps by those who live in the setting, to the point where some consider that it is up to each person to be able to create their waste dump in front of their house to avoid people being able to "throw away their waste at others". As one interviewee said, "We cannot continue to let people from elsewhere come and dump their waste in front of our businesses. We are tired. For example, this pile of waste is increasing every day. The most worrying thing is that some people come in surreptitiously at night to throw away anything. In any case, if I see someone pouring their rubbish here again, I will give him a hard time" [39].

To avoid conflicts, some people only throw away their (considerable) waste at night, without witnesses. This is why some waste piles often increase daily, without anyone seeing or knowing who is responsible. Informal waste heaps are created without the main initiators being known and usually start with one piece of waste (small empty packet of biscuits, old cloth, etc.) before becoming a large pile of garbage (such as the one in Figure 7). Formal and informal waste dumps can include such things as faeces, medicines, tyres, and even human or animal remains. The populations of Yaoundé with no access to regular waste collection services are forced to use their immediate environment to dispose of their waste, without considering the impact of these actions on their living environment. Furthermore, the Head of the Hygiene and Environment Department of YDM 6 states: "The unauthorised dumps are formed near water sources, or the waste is thrown into them leading to surface water and groundwater pollution through leachates. In addition, the

city's gutters are completely blocked because of the waste resulting from the incivility of citizens, who throw anything anywhere" [40].

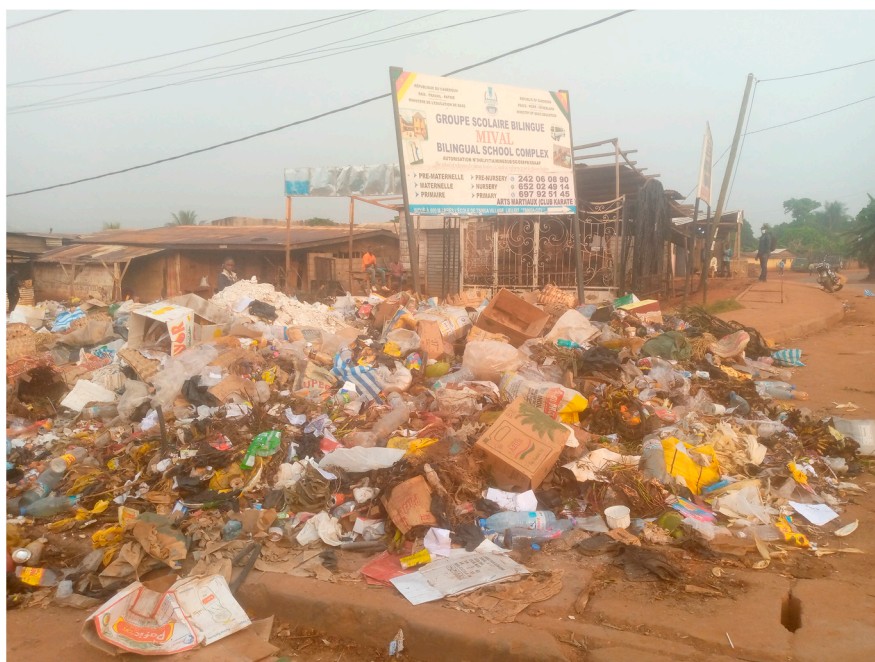

**Figure 7.** Informal depot straddling the road and pavement in Tsinga-City (YDM 1). Author's field photo, March 2023.

Often, the illegal incineration of waste is an informal waste treatment method used by people; this contributes significantly to the degradation of air quality. The incineration is done in waste containers or directly on the ground (or both) in neighbourhoods and areas both controlled by the public authorities or left unsupervised. In a context where waste sorting remains unknown, all types of waste are often burnt in the same place, without any knowledge of the environmental risks involved and with a substantial impact on the ozone layer (nitrogen dioxide, carbon oxide, nitrate oxide, etc.) [28].

Figure 8 shows a 6 m$^3$ waste container on fire (Figure 8). It shows waste being poured on the ground, which is also likely to be burnt. Waste management "is appalling in Yaoundé. But it leaves some inhabitants indifferent, including the authorities" [41]. The failure to collect or delays in waste collection around markets and homes and along public roads can be explained by the frequent delays in payment to the private companies in charge of waste collection and the lack of a budget to acquire the necessary equipment [42]. Constantly under budgetary pressure, the city's seven districts struggle to remove waste regularly and to effectively fight against the illegal dumps that multiply in their administrative district. As a result, "waste from markets, industries, hospitals, schools, and administrations mixes and increases the financial burden" [37]. The practical implementation of the orientation law on decentralization constitutes a significant challenge for sustainable development in Yaoundé [43].

Decentralisation, as a process of transferring competencies from the state to local authorities, aims at improving the regularity of waste removal and the living conditions of the population at all levels. According to Kuaté [44], the decentralisation policy is aimed at granting greater autonomy to public authorities vis-à-vis the central power. Although still at the infancy stage, Bliki [45] notes the potential financial problems of this policy, especially with regard to household waste management in sub-Saharan Africa. In the case of Yaoundé in particular, "the household waste management sector suffers from a serious lack of funding. In 2021, for example, the Ministry of Finance has allocated XAF 1,170,793,000 (EUR 1,784,862) of the Cameroonian state's share to the YCC (which also

contributes 1 billion per year). But with about 5 million inhabitants in the city, this is largely insufficient to cover household waste management" [36].

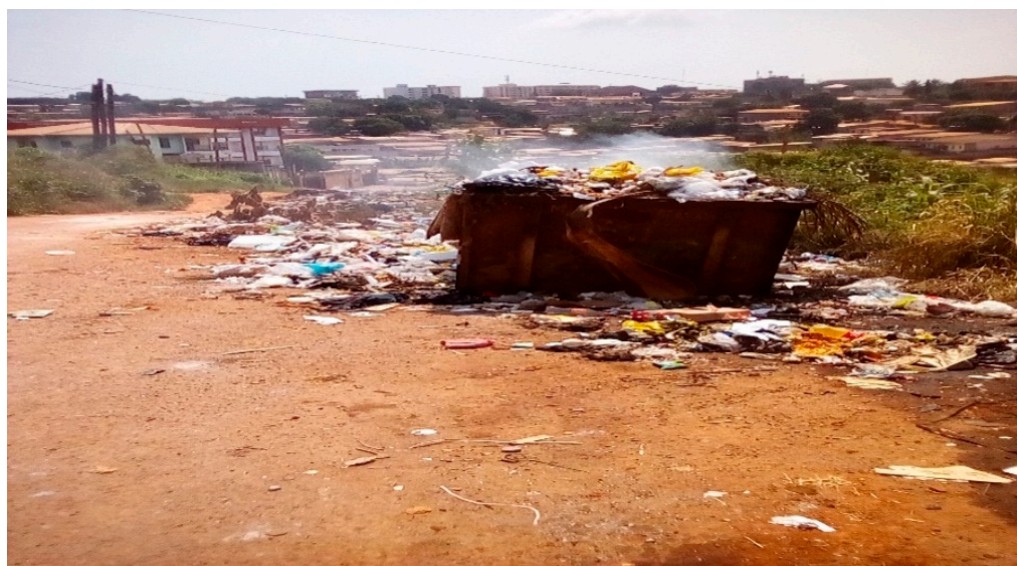

**Figure 8.** Open burning of waste at Eleveur in YDM 5. Author's field photo, November 2023.

In this respect, the need for more funding for household waste management is justified by the low level of financial contribution from the State in the face of growing waste production by an ever-increasing population. In concrete terms, the daily waste production per inhabitant in Yaoundé varies between 0.5 and 0.8 kg/day, with an average of 0.62 kg/day; this corresponded to the overall production of 1920 tonnes per day in 2016 [33]. This increased waste production has made it difficult for the YCC to manage waste disposal [37]. In September 2022, for example, the YCM distributed only 35,000 garbage bags to the 7 YDMs to be distributed in turn to the populations to boost the fight against insalubrity in Yaoundé. The sustainability of such actions is not guaranteed due to the lack of organisation and corruption in various forms [16].

*3.4. Direct Consequences of the Waste Crisis: Economy, Ecology, and Public Health*

From an economic point of view, the inferior quality of household waste management not only causes serious health problems for the public, who are the first victims, but also leads to unavoidable expenses for the state, which must contribute to treating the sick [26]. The populations of Yaoundé's precarious neighbourhoods spend a significant part of their income on the treatment of diseases caused by poor household waste management [27]. Malaria, yellow fever, typhoid, and cholera are among the common diseases affecting the Yaoundé population. Almost everyone interviewed reported suffering from one of these diseases in the three months prior to the interview.

Another consequence includes the flooding of roads, primarily due to clogging of drainage systems. The city centre (Poste Centrale), which covers barely 1% of the area, has repeatedly been affected by devastating floods caused by blocked drains, taking up "about 10% of the city's budget every year" [37]. In terms of public health, it has been shown that poor management of household waste leads to breeding grounds for disease vectors (mosquitoes, flies, etc.) [35]. In addition, waste is the leading cause of most of the endemic diseases that strike African cities [46]. Yaoundé appears to present significant health risks because it faces rampant and unplanned urbanisation in all respects [16]. For example, according to the World Health Organisation [47], between 80 and 85% of intertropical, waterborne, or vector-borne diseases are closely linked to poor sanitation. Most people interviewed for this study are socio-economically vulnerable; thus, the waste collection deficiency is concerning for local authorities and risk-aware people.

As per the urban ecology, at least 40% of the population of Yaoundé with no access to waste collection services use their immediate environment to dispose of their waste [38]. In working-class neighbourhoods (Melen, Étoug-Ébé, Nsam, Mokolo Elobi, Briqueterie, Tsinga, etc.), people dispose of their waste indiscriminately, to the point where unauthorised dumps pose serious environmental and health risks. In Yaoundé, waste officially collected throughout the city is dumped without adequate care in the locality of Nkolfoulou I, a small town on the border with Yaoundé. The population helplessly witnesses the pollution of the water and the contamination of the soil, which affects their main livelihoods, namely agriculture and fishing [19]. The waste collected needs to be treated, as in many cities [32,48]. However, it is buried in the ground and covered with soil as a (final) solution. There are no institutional recycling solutions. Nevertheless, some micro-recycling efforts are carried out by local environmental associations (CAFROMI, Tamtam-Mobile, etc.). These recycling efforts consist mainly of collecting, cleaning, and reselling PET and glass bottles to local businesses. The money raised funds waste sorting awareness campaigns and donations to vulnerable households (school supplies, medical treatment). Those associations deserve to be supported financially and materially for public recognition of recycling as contributing to the sustainable management of household waste [10].

## 4. Discussion and Conclusions

The issues of household waste management and sustainable development have been the subject of much research in the social sciences. Several authors and national and international organisations have analysed the issues of household waste management and sustainable development in developing countries in general and in Cameroon in particular. The World Bank and the United Nations Development Programme strongly recommend inclusive waste management through stakeholder involvement for sustainable household waste management [23,44]. However, these reports are mainly limited to the environmental dimension of household waste management and need to be more grounded in specific social realities. Sotamenou [24] demonstrates that the household waste management sector is a sector that can lead Cameroon toward socio-economic development. He proposes a progressive household waste management system that integrates pre-collection, composting, collection and transport, landfill, leachate treatment, and biogas operations. However, elements in addition to the economic dimension are needed to motivate and guarantee sustainable management of household waste [47].

This research aims to understand and explain the factors underlying the waste management crisis in Yaoundé, where the authorities constantly celebrate their commitment to the principles of sustainable development. It shows that the current quality of waste management is deficient, negatively influencing the environment, the urban economy, and public health. From a methodological point of view, the collection, processing, analysis, and interpretation of the data show that the household waste management in the area is insufficient to achieve the SDGs due to the weak will of the authorities and the public. In other words, the extent of informal waste dumping in the streets of Yaoundé can be explained by the lack of enforcement of existing sanctions and the many stakeholders involved in the interpretation of existing legal texts. Therefore, the research calls for adopting a mode of waste governance that values action and the population's well-being more than the simple ratification of international conventions as well as the production of local texts without an application on the field. With a high threat of disease transmission [49], the situation in Yaoundé is of concern, as it is in other cities in the global south where waste management remains disastrous despite regulations, with the state having taken little action to address the crisis.

Given that over 90% of the waste produced in Africa is disposed of in uncontrolled landfill sites and that the volume of waste is constantly increasing and is set to double by 2025 [50], this research adds to the literature in two major ways: first, mapping of the locations of waste disposal points in public spaces. The map shows that the shortcomings in the waste management system that were previously attributed exclusively to the public

authorities involve a dynamic process that also includes local people. In contrast to other studies conducted on the subject [41,46,51], this research shows that responsibility is collective and that sustainable solutions should involve public authorities and local people by coordinating bottom-up and top-down participatory and inclusive approaches. This is the sine qua non for public authorities to break out of the status quo that is undermining their efforts. Secondly, the combination of socio-anthropological and cartographic methods on the scale of a capital city has enabled us to see the extent of the risks to which urban populations are exposed. In this way, solutions can be envisaged according to the needs of the districts, depending on the critical situations in certain neighbourhoods. The latter have made it possible to understand the norms in force and their dynamics in household waste management in a local context (in line with the global), to account for the choices and decisions of actors, and to analyse the strategies developed by stakeholders to circumvent the norms. Yaoundé is stagnating under its waste. Other research related to household waste in Yaoundé could focus on the comparative analysis of specific management possibilities between the different communes of Yaoundé to assess the courses of action for improving sustainable social practices.

**Author Contributions:** Conceptualization, S.N. and A.F.T.; methodology, S.N., A.F.T. and J.-C.B.M.; validation, A.K. (Armel Kemajou), J.C., J.-C.B.M. and A.F.T.; formal analysis, J.C. and J.-C.B.M.; investigation, A.F.T. and A.K. (Abdou Kouomoun); resources, J.C.; writing—original draft preparation, S.N.; writing—review and editing, S.N., J.-C.B.M. and A.K. (Armel Kemajou); supervision, J.C. All authors have read and agreed to the published version of the manuscript.

**Funding:** This research received no external funding.

**Institutional Review Board Statement:** Not applicable.

**Informed Consent Statement:** Not applicable.

**Data Availability Statement:** No new data were created. The interviews conducted are primarily covered by a full non-disclosure agreement.

**Acknowledgments:** We are thankful to Armand Leka Essomba for their advice and to all who contributed to the success of this research.

**Conflicts of Interest:** The authors declare no conflict of interest.

## Appendix A

Table A1 specifies some of the international conventions to which the State of Cameroon adheres in the framework of its household waste management policy and whose objective is to protect the ecology and public health and promote economic growth.

**Table A1.** Some international conventions ratified by Cameroon regarding waste management.

| Convention | Signature Date | Objectives |
|---|---|---|
| Vienna Convention for the Protection of the Ozone Layer | 22 March 1985 | Promote appropriate measures to protect human health and the environment against adverse effects resulting or likely to result from human activities that modify or are likely to modify the ozone layer. |
| Basel Convention on the Control of Transboundary Movements of Wastes and their Disposal | 22 March 1989 | Ban on the import and export of hazardous waste |
| Montreal Protocol on Substances that Deplete the Ozone Layer | 30 August 1989 | Phase out the production of ozone-depleting substances (ODS) in order to reduce and protect their abundance in the atmosphere and the environment |
| United Nations Framework Convention on Climate Change | 19 October 1994 | Stabilise greenhouse gas concentrations in the atmosphere at a level that prevents dangerous anthropogenic interference with the climate system. |
| Bamako Convention on the Ban of the Importation of Hazardous Wastes and the Control of their Transboundary Movements in Africa | 20 March 1996 | To improve and ensure the environmentally sound management of hazardous waste, as well as cooperation between the African States involved |
| Stockholm Convention on Persistent Organic Pollutants | 23 May 2001 | Protection of human health and the environment from persistent organic pollutants. |

Source: Designed by Alirou Fit Touikoue and Salifou Ndam.

**Appendix B**

Table A2 shows the real political will of the Cameroonian State to regulate household waste management sustainably. Indeed, the objective of these legal and legislative texts is to define the different rights and duties of the stakeholders in charge of household waste management in order to guarantee regulatory assurance that meets the requirements of sustainable development in sanitation.

**Table A2.** Summary of the Cameroonian legal framework for waste management.

| Texts and Dates of Signature/Adoption | Area of Application | Objectives |
|---|---|---|
| - Law No. 76/372 of 2 September 1976 on the regulation of dangerous, unhealthy, or inconvenient establishments<br>- Law No. 98/015 of 14 July 1998 on establishments classified as dangerous, unhealthy, or inconvenient | Control of industries generating hazardous waste | To set the costs of inspection and control of dangerous, unhealthy or inconvenient establishments. |
| Law No. 89/027 of 29 December 1989 on toxic and hazardous waste | Hazardous and toxic waste management | Regulate the collection, transport, and treatment of hazardous and toxic waste. |
| Law No. 96/117 of 5 August 1996 on standardisation | Standardisation | Standardise waste management, similar to other areas of daily life. |
| Law No. 96/12 of 5 August 1996 on the framework law on environmental management (See Chapter IV: Articles 42, 43, 44, 45 and 46) | Environment | Regulate to ensure environmental protection. |
| - Law No. 2004/003 of 21 April 2004 governing urban planning in Cameroon<br>- Law No. 2004/018 of 22 July 2004 laying down the rules applicable to communes | Urban planning and management of municipalities | - Regulate urban development and construction throughout Cameroon.<br>- Establish rules for municipalities. |
| Law No. 74/23 of 5 December 1974 on the organisation of municipalities (see Article 71) | Waste management | To confer the municipality's new attributions in the field of collection, transport, and treatment of household waste. |
| - Decree No. 80/17 of 15 January 1980 setting the maximum rates for direct municipal taxes<br>- Law No. 2009/019 of 15 September 2009 on local taxation | Tax for the removal ofhousehold waste | - Indicate that the Tax for the Collection of Household Waste is included in the local development tax, which is itself based on salaries and the tax in full discharge.<br>- To set the rates for the Tax for the Collection of Household Waste and to examine the various budgets and administrative accounts of the two urban communities of Yaoundé and Douala (main cities in Cameroon). |
| Decree No. 2005/0577/PM of 23 February 2005 establishing the modalities for carrying out environmental impact studies | Environment and well-being | Impose and set the modalities for carrying out environmental impact assessments in all projects. |
| Order No. 01 of the Ministry of the Environment, Nature Protection and Sustainable Development of 15 October 2012 laying down the conditions for obtaining an environmental permit for waste management | Environment | Set conditions for obtaining environmental permits for waste management. |
| Joint Order No. 005 of the Ministry of the Environment, Nature Protection and Sustainable Development and the Ministry of Trade of 24 October 2012 laying down specific conditions for the management of electrical and electronic equipment and the disposal of waste from such equipment | Environment | Regulate the introduction of electrical and electronic equipment and the management of waste resulting from their use. |

**Table A2.** *Cont.*

| Texts and Dates of Signature/Adoption | Area of Application | Objectives |
|---|---|---|
| Joint Order No. 004 of the Ministry of the Environment, Nature Protection and Sustainable Development and the Ministry of Trade of 24 October 2012 regulating the manufacture, import, and marketing of non-biodegradable packaging | Environment, industry, and trade of biodegradable packaging | Prohibit the manufacture and marketing of non-biodegradable packaging in Cameroon, (those with a thickness of less than 60 μm). |
| Circular Note No. 8419/e/MINAT/DCPL/ SAA of 25 June 1979 on the national hygiene and sanitation campaign | Hygiene and sanitation | Regulate awareness via the National Hygiene and Sanitation Campaign. |
| Circular Note No. 069/NC/MSP/DMPHP/ SHPA of 20 August 1980 of the Ministry of Public Health on the collection, transport, and treatment of industrial waste, household waste, and sanitary sewage | Waste management and epidemic prevention | Regulate the collection, transport, and treatment of industrial waste, household waste, and sanitary waste and specify the technical conditions for waste disposal. |
| Circular note No. 0040/LC/MINAT/DCTD of 04 April 2000, of the Ministry of Territorial Administration and Decentralisation relating to the restoration of public hygiene and sanitation | Environment and sanitation | To fight against the deterioration of the general appearance of urban and rural areas. |

Source: Designed by Alirou Touikoue Fit and Salifou Ndam.

**Appendix C**

Table A3 presents five main groups of actors involved in household waste management in Yaoundé. They are categorised into three main groups, each with a specific role in waste management. For better waste management, these actors are intended to work in a chain (like a human organism). The dysfunction observed in one of the multiple attributions of each can affect the global chain.

**Table A3.** Stakeholders and their roles in the waste management process in Cameroon.

| Categories of Actors | Actors | Specific Responsibilities |
|---|---|---|
| Institutional or public actors | State of Cameroon | - Waste prevention<br>- Waste management<br>- Waste recovery<br>- Waste disposal<br>- Subsidising the collection, transport, and treatment of waste in cities |
| | Ministry of Housing and Urban development | - Responsible for the implementation of the national policy on housing and urban development<br>- Monitoring the application of hygiene and sanitation regulations and the collection and treatment of household waste |
| | Ministry of Environment, Nature Protection and Sustainable Development | - Development of a National Environmental Management Plan<br>- Examination of files relating to the disposal, recycling, and burial of waste, in liaison with the administrations concerned<br>- Periodic control of landfills |
| | Ministry of Territorial Administration and Decentralisation | - Household awareness in the neighbourhood<br>- Repression in case of infringements<br>- Financial contribution for waste collection<br>- Collection and treatment of household waste |
| | Ministry of Health Public | Prescription of the conditions for the removal and treatment of municipal waste. |
| | Ministry of Agriculture and Rural Development | Raising awareness of the population on the transformation of putrescible waste into fertilizer. |

**Table A3.** *Cont.*

| Categories of Actors | Actors | Specific Responsibilities |
|---|---|---|
| Institutional or public actors | Urban municipalities | - Hygiene and sanitation<br>- Raising awareness of the collection of the direct tax on household waste<br>- Technical control of solid waste collection, transport, and treatment |
| | District municipalities | To ensure sanitary, local, and environmental policing by means of technical sanitary officers |
| Non-governmental organisations (NGOs) | Civil society organisations (NGOs, neighbourhood associations, etc.) | Supporting the development of popular ecology movements and projects |
| Private companies and service concessionaires | Health and Safety of Cameroon (HYSACAM) | - Raising awareness among the population to avoid littering<br>- Collection, transport, and disposal of waste in cities |
| | Common Initiative Groups (CIGs), Small and Medium Companies (SMEs), etc. | - Private fee-based service for pre-collection of waste<br>- Action on pre-collection of waste |
| International donors | International Monetary Fund (IMF), World Bank, etc. | Provide financial means to support and sustain local waste management initiatives. |
| Users | Households | - Contribution to the financing of household waste disposal through taxation<br>- Direct financial contribution to the removal of waste from Common Initiative Groups (CIGs) and Small and Medium Enterprises (SMEs)<br>- Transport of waste from the place of production to the place of collection<br>- Obligation to clean the surroundings of their homes (decree of 24 May 2000) |

Source: Designed by Alirou Touikoue Fit and Salifou Ndam.

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
