# Peer review of "Urban Governance of Household Waste and Sustainable Development in Sub-Saharan Africa: A Study from Yaoundé (Cameroon)"

_waste, doi:10.3390/waste1030036_

Round 1
Reviewer 1 Report
This paper investigates the status quo of waste management in Yaoundé and tries to explain the factors underlying the current waste management, which would enrich our understanding about waste management in South-South countries. However, this paper still needs to be improved.
1. introduction. Authors should directly point out the research gap and the central research questions of this research. Meanwhile, the arrangement of this article should be supplemented.
2. as for the findings, in my opinion, authors should compare the differences of waste governance between Yaoundé and other regions (especially for the developed countries), and examine why the waste management in different regions presents heterogeneity. And some key information should be summarized in a Table or a figure, which would be more straightforward for the readership.
3. the conclusion part should be presented concisely with some concluding remarks which summarizes the critical findings of this study. And the same to the abstract.
4. Regarding the reference list, more academic journal articles should be included.
the language should also be polished.
Author Response
Thank you very much for your comment, it has been acknowledged, and we have majorly adjusted the manuscript to make it leaner and more structured.
Point 1: introduction. Authors should directly point out the research gap and the central research questions of this research. Meanwhile, the arrangement of this article should be supplemented.
Response 1: we have added two paragraphs supported by an extensive literature review to help focus the study and place it in a broader context.
Point 2: as for the findings, in my opinion, authors should compare the differences of waste governance between Yaoundé and other regions (especially for the developed countries) and examine why the waste management in different regions presents heterogeneity. And some key information should be summarized in a Table or a figure, which would be more straightforward for the readership.
Response 2: Lines 258-272 provide further information on the specificity of Yaoundé and the reality in other cities such as Dakar (Senegal) and Kigali (Rwanda). The case of Lausanne (Switzerland) (a study conducted by a co-author of this article) was cited for comparison. A comparative analysis table would have been interesting, but there is little precise data to enable us to carry out this exercise.
Point 3: the conclusion part should be presented concisely with some concluding remarks which summarizes the critical findings of this study. And the same to the abstract.
Response 3: the conclusion has been revised and the data in the abstract has been updated.
Point 4: Regarding the reference list, more academic journal articles should be included.
Response 4: More academic journal articles have been included.
Point 5: the language should also be polished.
Response 5: The language has been polished.
Reviewer 2 Report
The authors studied the household waste collection in the city of Yaounde, relying mostly on direct observations and on interviews with affected households, with agencies responsible for the waste collection and with other stakeholders. The topic is interesting, also for people living outside of Cameroon. The methodology that was to be followed is clear. However, the results and discussion section is not reflecting the structure of the methodology that is outlined in Figure 1. Specifically, it must be known what questions were asked in the interviews (both in the exploratory interviews in the first phase, and in the more detailed interviews in the second phase). While the answers collected can give both quantitative and qualitative information, it should be expected that quantitative answers are analyzed (also statistically) and tabulated. In the manuscript, as it is now, the reader finds more a narrative based on the authors' own observations to which are added selected qualitative statements coming from the interviews. This needs to be modified.
Nothing much said about the final resting place of the wastes as they are collected - this is of general interest to the reader and should be added. Are there any recycling efforts going on? Also, information of the frequency of collection and of typical quantities of wastes needs to be added.
Yaounde district 7 has no waste collection points. Is this correct?
Generally, there are some corrections that need to be done throughout.
In line 12 Yaounde is missing an "e"
The acronym SDG should be defined.
In line 46, it is "intelligibly"
In line 48 "its" should be replaced by "the country's"
The sentence in line 49 is not placed logically as there is no continuous flow of thought.
The sentence in lines 77-79 lacks clarity.
Author Response
Thank you very much for your comment, it has been acknowledged, and we have majorly adjusted the manuscript to make it leaner and more structured.
General comments
Point 1 : the results and discussion section is not reflecting the structure of the methodology that is outlined in Figure 1.
Response 1 : Generally, the structure of the results and the discussion have been revised in substance for a better understanding of the manuscript. The structure of the methodology is an overview of the steps followed by our research. The structure of the analysis, as it stands, is the result of a back-and-forth operation in the field.
Point 2: Specifically, it must be known what questions were asked in the interviews (both in the exploratory interviews in the first phase, and in the more detailed interviews in the second phase).
Response 2: see lines 139-141 and the following paragraphs.
Points 3: While the answers collected can give both quantitative and qualitative information, it should be expected that quantitative answers are analyzed (also statistically) and tabulated.
Response 3: See lines 257-260, 265-268, and 269-287.
Points 4: In the manuscript, as it is now, the reader finds more a narrative based on the authors' own observations to which are added selected qualitative statements coming from the interviews. This needs to be modified.
Response 4: The use of the socio-anthropological approach has oriented the current narrative model. The latter requires a detailed account of reality as it is on the field and not from the authors' point of view. In order to improve our manuscript as a whole, the analysis has been made more dense, and additional references have been added.
Point 5: Yaounde district 7 has no waste collection points. Is this correct?
Response 5: Additional waste disposal areas have been added to Yaounde 7 as well as the surrounding districts. The total Waste disposal areas have moved from 184 to 264. Furthermore, Fig. 3 to 5 have been updated with the latest data collected (7 May 2023). In the Abstract, lines 21-22 discussing the waste disposal geolocation were updated. Line 212-216 from point 3.1 was updated by integrating the latest data. Line 222-230 was also updated accordingly. Figures modified from Fig. 3 to 4 and from Fig 4 to 5.
2 -Comments on the quality of the English language
In line 12 Yaounde is missing an "e" = done
The acronym SDG should be defined = done
In line 46, it is "intelligibly" = done
In line 48 "its" should be replaced by "the country's" = done
The sentence in line 49 is not placed logically as there is no continuous flow of thought = corrected sentence
The sentence in lines 77-79 lacks clarity = corrected sentence

Round 2
Reviewer 1 Report
Dear authors,
Many thanks for your efforts on improving this manuscript. Yes, this manuscript has been improved on different aspects. However, it still keep some points to be determined.
Regarding the structure of this paper, a discussion would be helpful to highlight your innovational findings. For example, what's your critical findings, the reasons behind the heterogenous status quo of waste governance in comparison with other regions, and the policy implications to the authorities. similarly, "conclusion" part is needed, which can provide the readers with some take-away knowledge, in my opinion.
Also, based on the theoretical perspective, the authors mentioned they have utilized three different theories, including stakeholder theory, social constructivism, etc. Based on my own understanding, these theories are adopted to comprehend the waste governance in target city, as the analysis tools. Authors should emphasize more on the contribution on the waste management theory by telling a story about the urban waste governance in SSA. This is the marginal innovation/contribution to the theoretical landscape. And I highly recommend this part could be added in the Abstract. What's more, if possible, an analytical framework could be implemented in the Methodology.
Besides, please keep and double check the consistency of the format throughout the whole paper. (Line 396, Part 3.4).
Overall, this paper can be accepted after improving the above aspects.
The language use is acceptable, but a double-check in some terms would be highly recommended.
Author Response
Dear Reviewer
Thank you again for considering regards to our manuscript. Please see the list of comments and answers addressed by our team.
Point 1: Regarding the structure of this paper, a discussion would be helpful to highlight your innovational findings. For example, what's your critical findings, the reasons behind the heterogenous status quo of waste governance in comparison with other regions, and the policy implications to the authorities. similarly, "conclusion" part is needed, which can provide the readers with some take-away knowledge, in my opinion.
Response 1: Thank you very much for your comment. We have taken this comment into account in the discussion and conclusions. Instead of creating a special section for the discussion, we preferred to call the last part of our manuscript "discussion and conclusions".
Point 2: Also, based on the theoretical perspective, the authors mentioned they have utilized three different theories, including stakeholder theory, social constructivism, etc. Based on my own understanding, these theories are adopted to comprehend the waste governance in target city, as the analysis tools. Authors should emphasize more on the contribution on the waste management theory by telling a story about the urban waste governance in SSA. This is the marginal innovation/contribution to the theoretical landscape. And I highly recommend this part could be added in the Abstract. What's more, if possible, an analytical framework could be implemented in the Methodology.
Response 2: Thank you for this comment. The three theories (stakeholder theory, social constructivism, and strategic analysis) have been added in the Abstract and implemented at the end of the introduction (just before the methodology).
The other reviewer commented similarly, and we think we've considered that.
Point 3: Besides, please keep and double check the consistency of the format throughout the whole paper. (Line 396, Part 3.4).
Response 3: We have kept and double-checked the consistency of the format throughout the whole manuscript (including Line 396, Part 3.4). If necessary, we can use MDPI's English editing service for the final version of the manuscript.
Best wishes
Reviewer 2 Report
The revised version reads better than the original. Nevertheless, a formal set of questions in regard to a questionnaire is still missing and therefore it is very difficult for the reader to assess the information output. I also believe that some sort of statistical analysis for some questions should be performed.
The information on the preliminary waste disposal and collection in Yaounde is of interest to the reader. Equally of interest would be the [final] fate of the waste. Perhaps, the authors could elaborate on this in a paragraph or two.
In the conclusion, it says "From a theoretical point of view, contrary to other works which have mobilised either the theory of economic management [47] or cognitive approaches [22] or the idea of social representations [35]. This research has mobilised three different theories: the theory of social constructivism [44], the stakeholder theory [45], and strategic analysis [49]." This part should go into the introduction and should be more thoroughly explained.
Please note that during the correction/revision of the original manuscript, errors have occurred due to copy/paste actions. There are a number of fragmented sentences, sentences where normal nouns are capitalized in the middle of the sentence and there are also fragmented duplicated sentences (for instance in lines 230-233). Therefore, please proofread the edited version very carefully.
Author Response
Dear Reviewer
Thank you again for considering regards to our manuscript. Please see the list of comments and answers addressed by our team.
Point 1 : The revised version reads better than the original. Nevertheless, a formal set of questions in regard to a questionnaire is still missing and therefore it is very difficult for the reader to assess the information output. I also believe that some sort of statistical analysis for some questions should be performed.
Response 1: Thank you for this comment. We have reinforced the statistical analyses throughout the manuscript. This was necessary to make the manuscript easier to understand.
Point 2: The information on the preliminary waste disposal and collection in Yaounde is of interest to the reader. Equally of interest would be the [final] fate of the waste. Perhaps, the authors could elaborate on this in a paragraph or two.
Response 2: Many thanks for this comment. The information on the preliminary waste disposal and collection in Yaoundé has been elaborated on Line 567-671.
Points 3: In the conclusion, it says "From a theoretical point of view, contrary to other works which have mobilised either the theory of economic management [47] or cognitive approaches [22] or the idea of social representations [35]. This research has mobilised three different theories: the theory of social constructivism [44], the stakeholder theory [45], and strategic analysis [49]." This part should go into the introduction and should be more thoroughly explained.
Response 3: Many thanks for this comment. The three theories (stakeholder theory, social constructivism, and strategic analysis) have been added in the Abstract (see lines 24-26) and moved (and elaborated) in the introduction (see lines 160-173).
The other reviewer commented similarly, and we think we've considered that.
Points 4: Please note that during the correction/revision of the original manuscript, errors have occurred due to copy/paste actions. There are a number of fragmented sentences, sentences where normal nouns are capitalized in the middle of the sentence and there are also fragmented duplicated sentences (for instance in lines 230-233). Therefore, please proofread the edited version very carefully.
Response 4: We have kept and double-checked the consistency of the format throughout the whole manuscript (including Lines 230-233). If necessary, we can use MDPI's English editing service for the final version of the manuscript.
Best wishes
Round 3
Reviewer 2 Report
The authors have revised their manuscript.
Two things are missing:
Point 1 - the final disposal of the collected wastes in the bins is not clear. In lines 473-474 it does mention landfills - however, it would be great if the authors could find out about the fate of the wastes collected by the municipality. How much is incinerated - how much goes to a landfill? Is there recycling?
Point 2 - The authors should phrase out questions that had been posed in the questionnaire(s). Usually, it would be expected that such questionnaires are made available to the reader. It would be good, if the authors provide examples of questions posed.
Still, there are fragmented sentences:
line 73-75; lines 144-152; lines 245-247.
There are fragmented sentences so that the authors should take great care in proofreading their manuscript.
Author Response
Dear Reviewer
Many thanks you for considering regards to our manuscript. Please see the list of comments and answers addressed by our team.
Point 1 - the final disposal of the collected wastes in the bins is not clear. In lines 473-474 it does mention landfills - however, it would be great if the authors could find out about the fate of the wastes collected by the municipality. How much is incinerated - how much goes to a landfill? Is there recycling?
Response 1: Thank you for this comment. The waste collected by the municipality is not incinerated but buried at Nkolfoulou I (see lines 525-528). Further to your comment, we added: "There are no institutional recycling solutions. Nevertheless, some micro-recycling efforts are carried out by some local environmental associations (CAFROMI, Tamtam-Mobile, etc.). These recycling efforts consist mainly of collecting in the streets, cleaning, and reselling PET and glass bottles to local businesses. The money raised funds waste sorting awareness campaigns and donations to vulnerable households (school supplies, medical treatment). Those associations deserve to be supported financially and materially for public recognition of recycling as contributing to the sustainable management of household waste" (Line 548-560).
Point 2 - The authors should phrase out questions that had been posed in the questionnaire(s). Usually, it would be expected that such questionnaires are made available to the reader. It would be good, if the authors provide examples of questions posed.
Response 2: see lines 152-161: "Some of the questions raised include: How much waste do you produce daily? Where do you dispose of the waste produced or collected? Is it important for you to separate trash (why)? Is it possible for districts to regulate the amount of waste households produce? What impact does waste dumped on the streets have on your health and the environment? What direct action are you taking to ensure sustainable waste management? What are companies doing to reduce packaging? What happens to the waste collected (or dumped) on the streets?"
Point 3: Still, there are fragmented sentences
Responses 3:
-Lines 73-75: adjusted
-Lines 144-152: adjusted.
-Lines 245-247: adjusted.
We have also made adjustments in several parts of the manuscript.
Best regards